# STRUCTURED PACKING IN LLM TRAINING IMPROVES LONG CONTEXT UTILIZATION

## ABSTRACT

Recent advances in long-context Large Language Models (LCLMs) have generated significant interest, especially in applications such as querying scientific research papers. However, their potential is often limited by inadequate context utilization. We identify the absence of long-range semantic dependencies in typical training data as a primary hindrance. To address this, we delve into the benefits of frequently incorporating related documents into training inputs. Using the inherent directory structure of code data as a source of training examples, we demonstrate improvements in perplexity, even for tasks unrelated to coding. Building on these findings, but with a broader focus, we introduce Structured Packing for Long Context (SPLICE). SPLICE is an innovative method for creating training examples by using a retrieval method to collate the most mutually relevant documents into a single training context. Our results indicate that SPLICE enhances model performance and can be used to train large models to utilize long contexts better. We validate our results by training a large 3B model, showing both perplexity improvements and better long-context performance on downstream tasks.

## 1 INTRODUCTION

Large language models (Brown et al., 2020; Chowdhery et al., 2022; Lewkowycz et al., 2022; OpenAI, 2023; Bai et al., 2023) have completely transformed the field of AI and natural language processing. Long-context language models (LCLMs) are becoming increasingly popular and have the potential to unlock new capabilities (Anthropic, 2023). However, recent studies (Liu et al., 2023; Tworkowski et al., 2023) have highlighted numerous issues with their context utilization. Even if they have the technical ability to process long context, some models fail in practice, as evidenced by their poor performance even on synthetic retrieval tasks (Li et al., 2023a). Moreover, LLMs are often distracted from retrieving relevant information when provided with multiple documents in the prompt (Tworkowski et al., 2023; Liu et al., 2023; Shi et al., 2023), and typically struggle to utilize information from the middle of their input (Liu et al., 2023). There might be multiple reasons behind these issues, requiring structured research to address them.

In this work, we take a step towards better context utilization in LCLMs. We focus on training data, keeping other components (e.g., the architecture and training objectives) unchanged. The fundamental question then is *how to construct training examples such that using long context is beneficial for the model* (e.g., it allows for obtaining lower perplexity or acquiring new downstream capabilities) while being non-trivial (like simple copying). We show that including multiple related documents in a training context can be beneficial - not only for evaluating on multiple documents, but also when considering language modeling perplexity on long, naturally occurring, single documents, which our evaluation is aimed for. First, we showcase this using the natural directory structure of the code data. Following that, we propose SPLICE, Structured Packing for Long Context, that constructs training examples consisting of multiple similar documents selected using a retrieval method of choice (e.g., BM25).

We empirically validate SPLICE on a large-scale code generation dataset (Li et al., 2023b), demonstrating improved perplexity evaluated at larger contexts, as well as a higher performance on TREC (Li & Roth, 2002; Hovy et al., 2001), Qasper, and synthetic task from (Liu et al., 2023). For that, we use the dataset generated by our method to continue pretraining an OpenLLaMA 3B LLM using

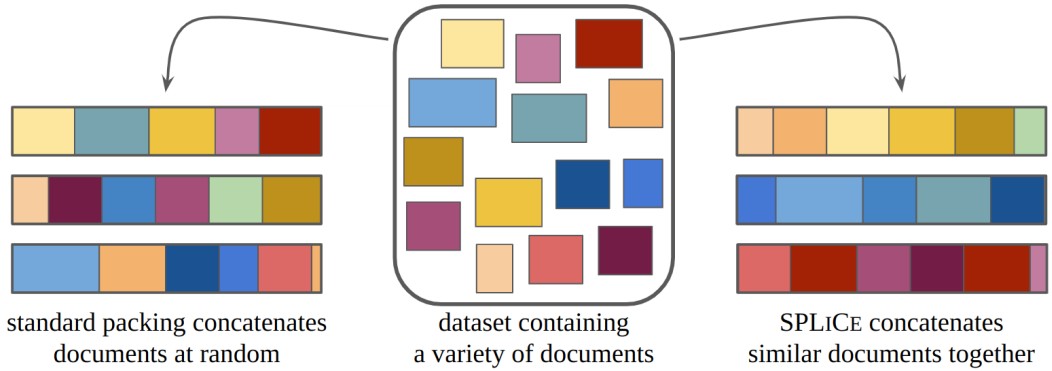

Figure 1: Training examples generated by the standard sampling procedure and proposed SPLiCE method.

the methodology described in (Tworkowski et al., 2023), resulting in a long-context language model with improved capabilities. Our contributions can be summarized as follows:

- We show that structuring training data is a viable way of improving the context utilization of LLMs. We validate this using the natural directory structure of code data.

- We introduce SPLiCE, a novel method of constructing training examples for long-context language models, which utilizes retrieval (implemented using BM25 or Contriever (Izacard et al., 2022)) to find relevant documents without relying on directory structure. We show that using SPLiCE during training on code matches the model's performance of the directory-structure method while being more generally applicable.

- We present a thorough analysis of the design choices of SPLiCE, the number of retrieved documents, and the order in which documents are merged into a training example.

- We validate SPLiCE by training a large-scale model, OpenLLaMA 3B v2 (Geng & Liu, 2023). We show that SPLiCE leads to improved perplexity on long-context evaluation, improved in-context learning ability and better retrieval performance.

## 2 METHOD

SPLiCE is a method of constructing the training examples that allows improvement of large-scale models' long-context utilization and, consequently, their performance on a range of tasks. Through experimental study, we positively validate that such an approach leads to better long-context capabilities.

**Rationale and intuitions** Capturing long-range dependencies is believed to enhance language modeling, e.g. as demonstrated in the study by (Borgeaud et al., 2022) which highlights the advantages of incorporating global dependencies in retrieval-augmented language models. However, in typical datasets, the occurrence of such dependencies is rare (de Vries, 2023) and diminishes as the distance between elements increases, leading to challenges in model learning. Our approach addresses this challenge by focusing on increasing dependency density, achieved by creating training examples that consist of multiple related documents. The rationale of this strategy is reinforced by the work of Levine et al. (2022), which shows that the trained model establishes stronger dependencies between text segments present in the same training example.

In our work, we study the following methods:

**Baseline** The standard approach, commonly used in LLM training pipelines (Brown et al., 2020), is to randomly sample documents from the dataset and concatenate them to make training examples of a desired context (also known as example packing).

**Repository-level code data** Certain datasets encompass meta-information regarding the data structure, which can be utilized to devise training examples. Take, for instance, a dataset con-

sisting of GitHub repositories (e.g., (Li et al., 2023b)): rather than grouping documents randomly into a single context, we might group documents based on their subdirectory in a repository, thereby crafting a training example with potentially more cross-document dependencies compared to those generated randomly. This methodology, elaborated in (Wu et al., 2022), led to notable perplexity enhancements with augmented context length, suggesting that data orchestrated in such a manner could exhibit valuable long-context training signals. In our study, we strive to quantify the advantages of this methodology against the random packing baseline outlined above while maintaining the set of training documents to eliminate potential confounders.

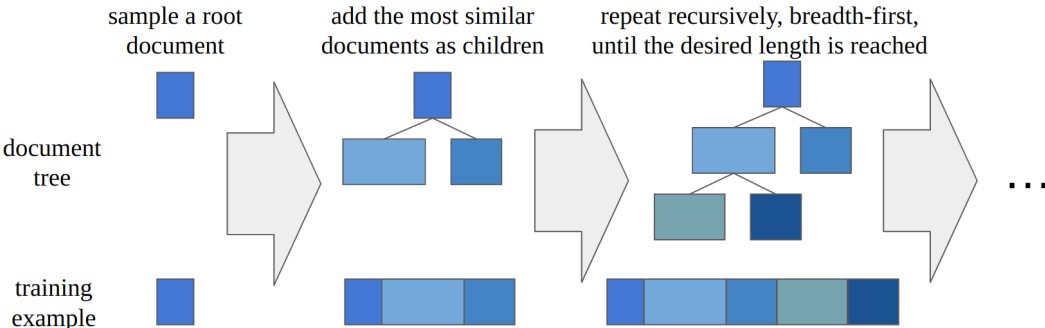

Figure 2: Visualization of SPLICE training example generation.

**Structured Packing for Long Context (SPLICE)**   The additional structure provided by meta-information is unavailable in most datasets. We propose to overcome this challenge by using a retrieval method to identify related documents. We test two choices of retrieval methods BM25 (Robertson & Zaragoza, 2009; Bassani, 2023) and Contriever-MSMARCO (Izacard et al., 2022)). With those tools at our disposal, we propose SPLICE, a method that creates training examples by building a tree of documents. It starts from a randomly sampled document and continues in a breadth-first manner, expanding each node with top-$k$ similar documents from the corpus. The final sequence consists of a flattened tree according to a certain traversal strategy (e.g., level-order traversal). An intuitive visualization of SPLICE is presented in Figures 1 and 2, while a detailed pseudocode is given in Algorithm 1. We note that the general formulation of SPLICE is flexible. For example, the parameter $k$ allows for interpolation between two modes of retrieval. For $k = 1$ SPLICE creates a path of related examples simulating a long document. For larger values of $k$ our method generates examples that are similar to the ones used by retrieval augmented models. Note that the dataset must be reasonably curated for SPLICE to work efficiently. For example, a low duplication rate incentivizes the model to utilize long context beyond simple copying.

## 3   EXPERIMENTS WITH MEDIUM-SCALE MODELS

In this section, we conduct a comprehensive examination of the impact of document packing into training examples on long-context performance. We demonstrate that leveraging the inherent repository structure of code data can yield notable perplexity enhancements. Additionally, we establish that our proposed method, SPLICE, either matches or surpasses this approach, exhibiting broader applicability to non-code data. We stress that we do not aim to surpass the approach to concatenate repository files together, instead we aim to show that SPLICE is more general and can be used for non-code data. We carry out an ablation study on SPLICE to scrutinize its design elements and hyperparameters in Appendix C.1. For this segment, due to computational constraints, we engage with 270M models to ensure a meticulous investigation. In Section 4, we affirm that these findings extend to large-scale models.

### 3.1   EXPERIMENTAL SETUP

To investigate the impact of input data organization on enabling language models to leverage extended context, we employ 270M models following a specific protocol. Initially, we pre-train a

---

**Algorithm 1** SPLICE training example construction

---

**Input:**
    $D$: the document corpus,
    $k$: the breadth hyper-parameter
    $L$: Maximum length of the returned training example
**Output:** training example consisting of concatenated documents
$d_r \sim D$                                                  ▷ Sample the root document
$D = D \setminus \{d_r\}$
$C = d_r$
$Q = \texttt{empty queue}$
$Q.\text{PUSH}(d_r)$
**while** $Q \neq \emptyset$ and $\texttt{len}(C) \leq L$ **do**
    $d = Q.\text{POP}()$
    $d_1, \ldots, d_k = \text{RETRIEVE}(d, k)$           ▷ Retrieve top $k$ most similar documents to $d$ using a
                                                     selected method, e.g., BM25
    **for** each $d_i$ in $d_1, \ldots, d_k$ **do**
        **if** $d_i \in D$ **then**             ▷ RETRIEVE uses a precomputed index, and may return
                                               documents that are already in $C$
                $C = C.\text{CONCAT}(d_i)$                   ▷ Append $d_i$ to $C$
                $Q.\text{PUSH}(d_i)$
                $D = D \setminus \{d_i\}$
**return** TRIM$(C, L)$

---

270M model on 6.3B tokens from the original RedPajama mixture (TogetherComputer, 2023), using a context size of 2K. Subsequently, we continue pretraining this model on 1B tokens with an extended context length of 32K, employing the Focused Transformer (FoT) objective Tworkowski et al. (2023), on a balanced mixture of the original RedPajama data (TogetherComputer, 2023) and long-context data assembled using the specified method. This approach is driven by practical factors: training with a shorter context expedites the process, and as demonstrated by (Chen et al., 2023; Tworkowski et al., 2023), the model training can be continued on longer documents from the original data source while maintaining the desired long-context perplexity scaling. During the long-context pretraining phase, only the latter half of the mixture is altered, retaining the original RedPajama portion intact. This approach is loosely inspired by (Ouyang et al., 2022) and (Rozière et al., 2023) and aims to prevent the model from overfitting to artificially created long documents. Only the effect of data organization into training examples is tested, keeping other factors like architecture intact. For a detailed description of the model architecture, refer to Appendix A, and for information regarding the training data, see Appendix F.

In the evaluation, we employ a context length of 32K and measure perplexity on held-out portions of the arXiv (Azerbayev et al., 2022) and StarCoder (Li et al., 2023b) datasets. The selection of these evaluation datasets is motivated by their inclusion of documents that stand to benefit from long-context information. Specifically, functions are often re-utilized, and similar terms are employed across papers. The advantage of long context in these datasets has been demonstrated for both arXiv (Chen et al., 2023) and StarCoder (Li et al., 2023b). We exclude documents with fewer than 32K tokens and truncate those exceeding this length. For further details regarding the chosen datasets and the evaluation setup, refer to Appendix F.

## 3.2 SPLICE ON CODE DATA

In this section, we compare SPLICE against the BASELINE and repository-level code data (REPO) packing methods. We test two retrieval approaches for SPLICE, the first one is BM25 (Robertson & Zaragoza, 2009; Bassani, 2023), the second one is and Contriever-MSMARCO (Izacard et al., 2022)) and denote them as SPLICE BM25 and SPLICE CONT respectively. In SPLICE BM25 retrieval takes into account the whole content of the document, whereas in SPLICE CONT we use the first 512 tokens and Faiss (Johnson et al., 2017b) for fast approximate inner-product search (for details see Appendix H). To make this comparison possible we use code data (required by REPO). We observe, see Table 1, that SPLICE significantly outperforms the BASELINE, and is either slightly better or

on par with REPO. This is a promising result, as SPLICE is more general and can be applied to non-code data (see Section 3.3).

Table 1: We train a 270M parameter model on a 50/50 mixture of the RedPajama data (organized in a standard way) and code data (organized in one of four ways: SPLICE BM25, SPLICE CONT, REPO, BASELINE – standard). We evaluate the perplexity of the models on arXiv and various subsets of StarCoder. The boldface denotes the best result within the training mixture. Note that SPLICE is generally on par or better than REPO and beats the BASELINE by a large margin. For detailed results, see Appendix B.

| Training Data | Method | arXiv | Code | | | | Code & arXiv |
|---|---|---|---|---|---|---|---|
| | | | Haskell | Python | CUDA | All | |
| C | SPLICE BM25 | **5.46** | **3.20** | **2.81** | **2.22** | **2.94** | **3.10** |
| | SPLICE CONT | 5.48 | 3.22 | 2.82 | 2.23 | 2.96 | 3.11 |
| | BASELINE | 5.55 | 3.37 | 2.93 | 2.33 | 3.07 | 3.23 |
| | REPO | 5.47 | 3.22 | 2.83 | 2.24 | 2.96 | 3.12 |
| C# | SPLICE BM25 | **5.52** | **3.33** | **2.90** | **2.46** | **3.11** | **3.26** |
| | SPLICE CONT | 5.53 | 3.35 | 2.91 | 2.48 | 3.12 | 3.27 |
| | BASELINE | 5.65 | 3.58 | 3.07 | 2.65 | 3.31 | 3.46 |
| | REPO | 5.53 | 3.35 | 2.91 | 2.49 | 3.12 | 3.27 |
| Python | SPLICE BM25 | **5.47** | **3.25** | **2.53** | **2.41** | **3.02** | **3.17** |
| | SPLICE CONT | 5.49 | 3.28 | **2.53** | 2.43 | 3.03 | 3.19 |
| | BASELINE | 5.57 | 3.46 | 2.62 | 2.56 | 3.17 | 3.32 |
| | REPO | 5.48 | 3.27 | 2.54 | 2.44 | 3.03 | 3.18 |

To deepen the analysis, we prepared three subsets of the C code and ran the tested methods on each of them. In Table 2, we report the mean and standard deviation. We observe that the differences between the subsets are minimal, which indicates training stability and confirms the statistical significance of our results. For details about how the training sets were prepared please refer to Appendix F.

Table 2: Perplexity for training on a $50/50$ data mixture of RedPajama and C code. For the C code we use three different subsets and report the mean and standard deviation.

| Training Data | Method | arXiv | Code | | Code & arXiv |
|---|---|---|---|---|---|
| | | | Python | All | |
| C | SPLICE BM25 | **5.463** ± 0.002 | **2.810** ± 0.002 | **2.942** ± 0.005 | **3.100** ± 0.004 |
| | SPLICE CONT | 5.477 ± 0.005 | 2.824 ±0.001 | 2.957 ± 0.006 | 3.115 ± 0.006 |
| | BASELINE | 5.550 ± 0.002 | 2.931 ± 0.008 | 3.073 ± 0.006 | 3.228 ± 0.005 |
| | REPO | 5.474 ± 0.007 | 2.827 ± 0.006 | 2.958 ± 0.009 | 3.115 ± 0.009 |

We also test our method with 64K context length, and report the results Table 16 in Appendix J.

## 3.3 SPLICE ON WEBTEXT DATA

The major advantage of our method over REPO is that SPLICE can be applied to any textual data. Table 3 shows that using our method the model outperforms the baseline when trained on a $50/50$ mixture of RedPajama (packed in the standard way) and Wikipedia/StackExchange (prepared using SPLICE). Notably, we observe that the model trained on StackExchange data significantly outperforms the one trained on C in the arXiv evaluation. Despite the non-code nature of StackExchange we still note perplexity improvements on code. Unsurprisingly, the perplexity on code is not as good as when training directly on code data. We conjecture that better data mixtures could bring further benefits.

Table 3: Perplexity results for training on a 50/50 data mixture of RedPajama and one of {StackExchange,C} organized using mentioned methods. Note that when evaluating on arXiv the model trained with SPLICE organized StackExchange outperforms the one trained on C.

| Training data | Method | arXiv | Code | | | | Code & arXiv |
|---|---|---|---|---|---|---|---|
| | | | Haskell | Python | CUDA | All | |
| Wikipedia | SPLICE BM25 | **5.64** | **3.82** | **3.26** | **2.87** | **3.55** | **3.68** |
| | SPLICE CONT | 5.65 | 3.87 | 3.30 | 2.92 | 3.59 | 3.72 |
| | BASELINE | 5.73 | 3.97 | 3.37 | 3.00 | 3.68 | 3.81 |
| StackExchange | SPLICE BM25 | **5.07** | **3.88** | **3.32** | **2.89** | **3.60** | **3.69** |
| | SPLICE CONT | 5.09 | 3.91 | 3.35 | 2.93 | 3.63 | 3.73 |
| | BASELINE | 5.14 | 3.94 | 3.36 | 2.94 | 3.65 | 3.74 |
| C | SPLICE BM25 | **5.46** | **3.20** | **2.81** | **2.22** | **2.94** | **3.10** |
| | SPLICE CONT | 5.48 | 3.22 | 2.82 | 2.23 | 2.96 | 3.11 |
| | BASELINE | 5.55 | 3.37 | 2.93 | 2.33 | 3.07 | 3.23 |

## 3.4 SPLICE WITH DIFFERENT CONTEXT EXTENSION APPROACHES

Our main experiments use the Focused Transformer (FoT) approach (Tworkowski et al., 2023) for context extension. FoT extends the context only in a couple of attention layers and does not change the positional embeddings of the remaining layers. This allows for computationally efficient long-context fine-tuning. To show that our method works more generally, we have also evaluated it using more standard approaches. That is, we checked the effectiveness of the method when context extension is done in all layers, and parameters of Rotary Positional Encodings are either adjusted as in CodeLlama (Rozière et al., 2023), adjusted with YaRN (Peng et al., 2023), or left without changes. Table 4 shows the results.

Table 4: Perplexity results for training on a 50/50 data mixture of RedPajama, and C# organized using the mentioned methods. We check how much SPLICE can help when fine-tuning for the larger context using methods different than FoT (Tworkowski et al., 2023). That is, we test the effectiveness of SPLICE when the context is extended to $16K$ in all layers and parameters of positional embedding are either adjusted as in CodeLlama (Rozière et al., 2023), with YaRN (Peng et al., 2023), or left without changes. The results indicate that SPLICE still brings perplexity improvements in all of these setups. For YaRN we have used the parameters from (Peng et al., 2023) and adjusted the scale factor appropriately.

| RoPe scale method | Training data | Method | arXiv | Code | | | | Code & arXiv |
|---|---|---|---|---|---|---|---|---|
| | | | | Haskell | Python | CUDA | All | |
| Naive | C# | SPLICE BM25 | **6.25** | **4.84** | **3.55** | **2.84** | **3.72** | **3.88** |
| | | BASELINE | 6.33 | 5.03 | 3.66 | 2.96 | 3.85 | 4.00 |
| | | REPO | **6.25** | 4.87 | 3.56 | 2.85 | 3.74 | 3.89 |
| CodeLlama | C# | SPLICE BM25 | **5.74** | **4.28** | **3.22** | **2.53** | **3.34** | **3.49** |
| | | BASELINE | 5.76 | 4.37 | 3.27 | 2.58 | 3.40 | 3.55 |
| | | REPO | **5.74** | **4.28** | **3.22** | 2.54 | 3.35 | 3.50 |
| YaRN | C# | SPLICE BM25 | **5.77** | **4.32** | **3.24** | **2.55** | **3.37** | **3.52** |
| | | BASELINE | 5.79 | 4.42 | 3.29 | 2.61 | 3.44 | 3.58 |
| | | REPO | **5.77** | **4.32** | **3.24** | 2.56 | 3.38 | 3.53 |

## 4 LARGE-SCALE MODELS

In this section, we show that SPLICE can improve the long context performance of large-scale language models. To this end, we use 3B parameter models and tasks that test in-context learn-

ing, question answering, and information retrieval capabilities. We provide perplexity results in Appendix E and additional experiments with a larger 7B parameter model in Appendix I

## 4.1 EXPERIMENTAL SETUP

We continue training OpenLLaMA 3B v2 (Geng & Liu, 2023; Geng, 2023) on the dataset defined in Section 3.1 - that is, a 50/50 mixture of RedPajama prepared in a standard way and C prepared using a method of choice. Specifically, we train two models with $32K$ context length for the same number of tokens, 5.4B, which is 21k training steps, on two datasets: one generated using SPLICE, and the other one using the BASELINE, see Section 2 for definitions. We use a batch size of 256K tokens per step, and learning rate of $1.5\mathrm{e}-5$ with linear warmup and cosine decay, following (Geng & Liu, 2023). We evaluate the perplexity on a mixture of long documents sourced from CommonCrawl, arXiv, and StarCoder (Li et al., 2023b). Otherwise, we follow the protocol described in Section 3.1.

### 4.1.1 IN CONTEXT LEARNING

To evaluate the improvements of the model's in-context learning ability we use TREC (Li & Roth, 2002; Hovy et al., 2001) question classification task. We provide the SPLICE trained model with $190$ to $1560$ in-context examples and compare the prediction accuracy against the BASELINE model. We sample the few-shot examples using 50 different random seeds and show the histogram of accuracy improvements in Figure 3 and additional results in Table 4 and Figure 7.

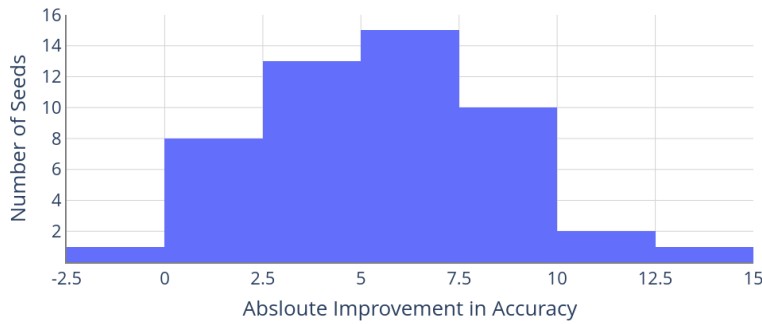

Figure 3: Histogram of accuracy improvement of SPLICE BM25 over BASELINE on TREC question classification task. The results are obtained by comparing the accuracy on the test set of TREC of the 3B model trained with SPLICE to the model trained with default data preparation method (BASELINE) across $50$ sets of in-context examples. Each set of in-context examples consists of $1560$ elements (roughly 32k tokens) randomly sampled (without replacement) from the training subset of TREC.

## 4.2 STANDARDIZED COMPARISON OVER LONG LANGUAGE SEQUENCES

We show that SPLICE can improve model performance on Qasper (Dasigi et al., 2021; Shaham et al., 2022). Qasper tests the model's ability to answer questions regarding research papers provided as a part of the input. The evaluated models were trained on a mixture of RedPajama prepared in a standard way and C prepared either in a standard way (BASELINE) or using SPLICE. Note that despite SPLICE using only code data here, we still see improvements in score on non-code, see Table 5.

## 4.3 KEY RETRIEVAL PERFORMANCE

Recent works indicated numerous issues related to long-context utilization. In particular, (Liu et al., 2023, Figure 7) proposed a benchmark retrieval task to measure the ability of language models to retrieve information from a long context. Specifically, the model is presented with a list of key-value pairs, which are 128-bit UUIDs, and is tasked to retrieve the value for a given key. Even though very simple, the task proved to be challenging for many open-source language models.

Table 5: Zero-shot performance on the validation subset of Qasper (Dasigi et al., 2021) and few-shot performance on TREC. For Qasper we use the implementation from Language Model Evaluation Harness (Gao et al., 2021). In the Harness implementation, yes/no questions are evaluated separately from open questions. Note that despite using SPLiCe for code data only we still have improvements in non-code tasks. For TREC we average results across 50 sets of in-context examples. Note that for TREC SPLiCe model is better for almost all choices of in-context examples (see Figures 3 and 7).

| Task | Context length | 3B Baseline | 3B SPLiCe |
|------|:---:|:---:|:---:|
| TREC | 32K | 73.9 ±3.9 | 79.3 ± 2.9 |
|      | 16K | 68.9 ±5.9 | 76.9 ± 3.1 |
|      | 8K  | 66.5 ±6.2 | 75.8 ± 3.5 |
|      | 4K  | 65.1 ±6.1 | 72.8 ± 4.9 |
| Qasper F1 | 32K | 23.1 | 23.9 |
|           | 4K  | 18.6 | 18.7 |

In Figure 4 we present the key retrieval performance of the 3B models trained with SPLiCe and Baseline. The dictionary contains 75 pairs, which is about 6k tokens. The performance depends heavily on the key position (which is also the case for other models, as indicated in (Liu et al., 2023)). The keys near the beginning are much harder to retrieve, and SPLiCe significantly improves the accuracy.

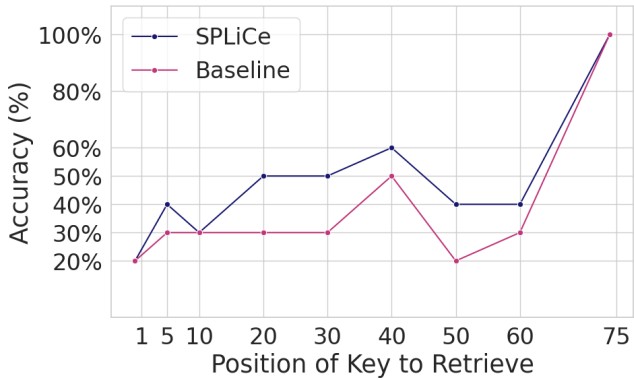

Figure 4: Key-value retrieval performance on a dictionary of 75 key-value pairs ($\approx$ 6k tokens). The 3B model trained with SPLiCe achieves much higher accuracy on hard-to-retrieve positions. Both models achieve 100% accuracy near the end because the desired key is in the local context (below 2048 tokens). The details about this task can be found in Appendix D.

## 5 RELATED WORK

Some recent studies have focused on enhancing training examples to improve the performance of language models. For instance, (Levine et al., 2022) developed a theory and demonstrated empirically that incorporating non-adjacent but semantically related sentences in training examples leads to better sentence embeddings and improves open-domain question-answering performance. Another work (Gu et al., 2023) introduced a pretraining framework grounded on the idea that text documents often include "intrinsic tasks". They showed that this approach substantially boosts in-context learning. Additionally, there is existing work on training long-context language models using repository-level code data, such as (Wu et al., 2022).

Our methodology diverges from these works in several key ways. Firstly, we focus on the document-level context during the training phase, as opposed to sentence-level (Levine et al., 2022) or paragraph-level (Gu et al., 2023) granularity. We demonstrate the efficacy of this approach in large-scale language modeling, specifically with OpenLLaMA 3B. Secondly, we construct a tree structure of related documents through BM25/Contriever-MSMARCO retrieval and linearize this structure to

form long-context examples. This allows for greater control over example coherence compared to relying solely on natural data structures like repository-level code.

## 6   LIMITATIONS AND FUTURE WORK

We show that structuring the training data is a viable way of improving the model's performance. The presented method can viewed as a general framework for organizing the documents into training examples (i.e. to achieve the repository-level grouping, it suffices to replace the `RETRIEVE`$(d, k)$ method with one that returns files from the closest catalog in the repository). This opens multiple further research avenues.

In this work, SPLICE constructed training examples by concatenating most matching documents (according to BM25/Contriever-MSMARCO). However, recent results of (Tworkowski et al., 2023) show that introducing unrelated data to the context may help the model learn better representations. We leave the study of how the choice of retriever (in particular, how the ratio of related and unrelated documents) affects performance for future work. Note that as BM25 is a text-statistic based retriever, it is not guaranteed that the documents returned by it are semantically related. Moreover, we have not tuned Contriever-MSMARCO, and it was provided only with a $512$ token long prefix of a document.

Another, avenue for future work is to study the granularity of the pieces from which the training examples are constructed. In this work, we focus on the document-level granularity. However, it is possible to construct training examples from smaller pieces, such as paragraphs or sentences.

Our method requires further studies to understand its scaling properties beyond the range presented in the paper. In particular, we leave scaling the context length to hundreds of thousands of tokens and experiments with models having more than 3B parameters for future work.

Our models were trained for a long context using the methodology presented in (Tworkowski et al., 2023). This methodology is similar to the approach used in (Wu et al., 2022) and allows extension of the model context with relatively small (compared to other methods) computational cost. Briefly speaking, it boils down to choosing a subset of attention layers for the context extension. We have additionally tested three popular context extension methods on a medium-scale (Naive, YaRN, and CodeLlama). We leave studies of SPLICE with other context extension methods as future work.

One of the approaches to training long-context language models is to use long conversational data (Li et al., 2023a). This approach is complementary to our method. SPLICE can utilize data that already exists in vast quantities and can be easily applied to different types of text (like code, Wikipedia articles, StackExchange questions, and answers, etc.) to further increase the number of long-context examples. We leave researching how SPLICE integrates with other methods for preparing the long-context data as future work.

In order to prepare the training data SPLICE uses each document exactly once (we mask out each used document for further retrieval). However, it is possible that allowing some high-quality documents to occur more than once may be beneficial.

Using highly correlated samples has the potential to result in training instability. However, during our experiments with StarCoder, Wikipedia, and StackExchange data we have noted no performance degradation. We leave the study of how SPLICE integrates with different types of data for the future. In particular, in our studies used datasets were reasonably deduplicated.

## 7   CONCLUSIONS

In this work, we present SPLICE, a novel method of constructing training examples for long-context language models, which utilizes BM25/Contriever-MSMARCO to find relevant documents and feed them to the model in a structured manner. We show that SPLICE improves the perplexity of on language modeling task and improves the performance on downstream tasks. We further show that SPLICE can be used to train large-scale models, resulting in a model with improved long-context utilization. Besides its direct usefulness, we believe that our work indicates multiple interesting research directions on how to improve the performance of long-context language models with structured data.

## 8    REPRODUCIBILITY

To ensure the reproducibility of our results we provide the details about how the data was prepared in Appendix F, the pseudocode of our method in Algorithm 1 and details about the trained models in Appendix A. We also attach the source code in the supplementary material.

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

## A  ARCHITECTURE

The architecture of our models is based on LLaMA (Touvron et al., 2023), and the architectural details can be found in Table 6. Briefly speaking, our architecture is similar to the one introduced in (Vaswani et al., 2017) with a few standard changes. First, we use only the decoder without the encoder part. Secondly, we perform RMSNorm before the input of both the attention and feed-forward modules. Thirdly, we use the LLaMA FeedForward module. Additionally, we use Rotary Position Embedding (Su et al., 2021). For context extension, we use Focused Transformer (Tworkowski et al., 2023). Table 7 presents the details about both standard and long-context pretraining.

Table 6: Architecture details. Focused Transformer context extension is applied in continued pretraining for 32K context and evaluation.

| Parameter/Model Size | 270M | 3B | 7B |
|---|---|---|---|
| Vocab Size | 32000 | 32000 | 32000 |
| Embedding Size | 1024 | 3200 | 4096 |
| Num Attention Layers | 12 | 26 | 32 |
| Num Attention Heads | 8 | 32 | 32 |
| Head Size | 128 | 100 | 128 |
| MLP Hidden Size | 4096 | 8640 | 11008 |
| FoT Context Extension Layers | [6, 9] | [6, 12, 18] | [8, 16, 24] |

Table 7: Training details. We pretrain a custom 270M parameter model and take a pretrained 3B/7B parameter OpenLLaMAv2 model (Geng, 2023).

| Stage | Parameter/Model Size | 270M | 3B | 7B |
|---|---|---|---|---|
| Pretraining | Context | 2K | 2K | 2K |
| | Tokens | 6.3B | 1T | 1T |
| Long Context Pretraining | Context | 32K | 32K | 32K |
| | Batch Size | 128 | 128 | 128 |
| | Start Learning Rate | 5e-5 | 1.5e-5 | 1.5e-5 |
| | End Learning Rate | 5e-6 | 1.5e-6 | 1.5e-6 |
| | Warmup Steps | 250 | 1000 | 1000 |
| | Tokens | 1B | 5.4B | 2B |

## B  DETAILED RESULTS

In this section, we present detailed results on all considered datasets. The training methodology is described in Section 3.1. Details about the number of tokens used for evaluation can be found in Table 14.

Tables 8 and 9 show the results of training 270M parameter model for 32K context on a 50/50 mixture of RedPajama data (organized in a standard way) and code data organized using a specified method. Table 8 contains detailed results from training on C# and Python. Table 9 contains results on C averaged across three different subsets of C (for details about the construction of those subsets see Appendix F). Both tables show that SPLICE outperforms the BASELINE by a significant margin and is slightly better than or on par with REPO.

The main advantage of SPLICE over the REPO approach is that it can be used also for non-structured data. Table 10 shows the detailed results of applying the SPLICE on non-code data. Note that training on non-code data allows us to improve the model perplexity on the arXiv dataset in comparison to the model trained on code.

Table 11 shows that a simple artificial extension of example length via random concatenation of documents does not help.

Table 8: Perplexity results comparing different ways of organizing the same data. All runs started from the same $270M$ model with 2048 context and were trained for 32K context on a 50/50 mixture of RedPajama (organized in a standard way) and code organized in mentioned ways. For details about training please refer to Section 3.1.

| | Altered Train Data: C# | | | |
| Eval/Method | SPLiCE BM25 | SPLiCE Cont | BASELINE | REPO |
| --- | --- | --- | --- | --- |
| ArXiv | **5.52** | 5.53 | 5.65 | 5.53 |
| C | **2.39** | 2.40 | 2.50 | 2.40 |
| C++ | **2.60** | 2.61 | 2.74 | 2.62 |
| CUDA | **2.46** | 2.48 | 2.65 | 2.49 |
| C# | **1.82** | **1.82** | 1.90 | **1.82** |
| Common Lisp | **3.41** | 3.44 | 3.72 | 3.42 |
| Dart | **2.14** | 2.16 | 2.31 | 2.16 |
| Emacs Lisp | **8.41** | 8.46 | 8.85 | **8.41** |
| Erlang | **2.80** | **2.80** | 2.95 | 2.81 |
| Fortran | **3.68** | 3.71 | 4.05 | 3.72 |
| Go | **1.94** | 1.95 | 2.06 | 1.96 |
| Groovy | **3.01** | 3.03 | 3.25 | 3.04 |
| Haskell | **3.33** | 3.35 | 3.58 | 3.35 |
| Java | **2.09** | 2.10 | 2.22 | 2.10 |
| Pascal | 3.61 | 3.62 | 3.80 | **3.60** |
| Python | **2.90** | 2.91 | 3.07 | 2.91 |
| Mean | **3.26** | 3.27 | 3.46 | 3.27 |

| | Altered Train Data: Python | | | |
| Eval/Method | SPLiCE BM25 | SPLiCE Cont | BASELINE | REPO |
| --- | --- | --- | --- | --- |
| ArXiv | **5.47** | 5.49 | 5.57 | 5.48 |
| C | **2.38** | 2.39 | 2.45 | 2.39 |
| C++ | **2.61** | 2.63 | 2.71 | 2.63 |
| CUDA | **2.41** | 2.43 | 2.56 | 2.44 |
| C# | **1.98** | 1.99 | 2.06 | 2.00 |
| Common Lisp | **3.23** | 3.28 | 3.47 | 3.27 |
| Dart | **2.15** | 2.17 | 2.27 | 2.17 |
| Emacs Lisp | 7.94 | 7.98 | 8.28 | **7.93** |
| Erlang | **2.70** | 2.71 | 2.82 | 2.71 |
| Fortran | **3.42** | 3.46 | 3.72 | 3.46 |
| Go | **1.95** | 1.96 | 2.03 | 1.96 |
| Groovy | **2.97** | 2.99 | 3.13 | 2.99 |
| Haskell | **3.25** | 3.28 | 3.46 | 3.27 |
| Java | **2.12** | **2.12** | 2.20 | 2.13 |
| Pascal | **3.57** | 3.58 | 3.73 | 3.58 |
| Python | **2.53** | **2.53** | 2.62 | 2.54 |
| Mean | **3.17** | 3.19 | 3.32 | 3.18 |

Table 9: To check the statistical significance of our results we prepare three subsets of C (details in Appendix F and train the models on a 50/50 mixture of RedPajama data (organized in standard way) and C data organized using one of the methods. Note that the standard deviation is much lower than the perplexity improvements from using SPLICE.

| Eval/Method | Altered Train Data: C | | | |
| | SPLICE BM25 | SPLICE CONT | BASELINE | REPO |
| --- | --- | --- | --- | --- |
| ArXiv | **5.463** $\pm$ 0.002 | 5.477 $\pm$ 0.005 | 5.550 $\pm$ 0.002 | 5.474 $\pm$ 0.007 |
| C | **2.126** $\pm$ 0.020 | 2.134 $\pm$ 0.020 | 2.173 $\pm$ 0.023 | 2.135 $\pm$ 0.020 |
| C++ | **2.396** $\pm$ 0.004 | 2.403 $\pm$ 0.002 | 2.467 $\pm$ 0.001 | 2.403 $\pm$ 0.006 |
| CUDA | **2.219** $\pm$ 0.002 | 2.235 $\pm$ 0.005 | 2.330 $\pm$ 0.009 | 2.239 $\pm$ 0.004 |
| C# | **1.939** $\pm$ 0.000 | 1.948 $\pm$ 0.001 | 2.016 $\pm$ 0.004 | 1.949 $\pm$ 0.002 |
| Common Lisp | **2.994** $\pm$ 0.072 | 3.042 $\pm$ 0.091 | 3.195 $\pm$ 0.078 | 3.043 $\pm$ 0.102 |
| Dart | **2.141** $\pm$ 0.002 | 2.155 $\pm$ 0.004 | 2.268 $\pm$ 0.009 | 2.156 $\pm$ 0.002 |
| Emacs Lisp | 7.857 $\pm$ 0.017 | 7.851 $\pm$ 0.020 | 8.098 $\pm$ 0.027 | **7.840** $\pm$ 0.019 |
| Erlang | **2.665** $\pm$ 0.002 | 2.680 $\pm$ 0.003 | 2.769 $\pm$ 0.007 | 2.676 $\pm$ 0.003 |
| Fortran | **3.306** $\pm$ 0.010 | 3.335 $\pm$ 0.010 | 3.554 $\pm$ 0.010 | 3.333 $\pm$ 0.012 |
| Go | **1.910** $\pm$ 0.001 | 1.924 $\pm$ 0.007 | 1.999 $\pm$ 0.002 | 1.924 $\pm$ 0.006 |
| Groovy | **3.009** $\pm$ 0.001 | 3.026 $\pm$ 0.006 | 3.147 $\pm$ 0.013 | 3.025 $\pm$ 0.007 |
| Haskell | **3.198** $\pm$ 0.001 | 3.221 $\pm$ 0.001 | 3.371 $\pm$ 0.008 | 3.220 $\pm$ 0.008 |
| Java | **2.075** $\pm$ 0.002 | 2.086 $\pm$ 0.001 | 2.161 $\pm$ 0.006 | 2.085 $\pm$ 0.005 |
| Pascal | **3.492** $\pm$ 0.026 | 3.496 $\pm$ 0.019 | 3.622 $\pm$ 0.021 | 3.513 $\pm$ 0.014 |
| Python | **2.810** $\pm$ 0.002 | 2.824 $\pm$ 0.001 | 2.931 $\pm$ 0.008 | 2.827 $\pm$ 0.006 |
| Mean | **3.100** $\pm$ 0.004 | 3.115 $\pm$ 0.006 | 3.228 $\pm$ 0.005 | 3.115 $\pm$ 0.009 |

Table 10: Perplexity results comparing different ways of organizing the same data. All runs started from the same $270M$ model with 2048 context and were trained for 32K context on a 50/50 mixture of RedPajama (organized in a standard way) and other data organized using one of the methods. For details about training please refer to Section 3.1. Note that the model trained with SPLICE on StackExchange outperforms the one trained on code on arXiv evaluation, showing the benefits of SPLICE's applicability to non-code data.

| | Altered Train Data: StackExchange | | |
|---|---|---|---|
| Eval/Method | SPLICE BM25 | SPLICE CONT | BASELINE |
| ArXiv | **5.07** | 5.09 | 5.14 |
| C | **2.68** | 2.69 | 2.70 |
| C++ | **3.02** | 3.04 | 3.06 |
| CUDA | **2.89** | 2.93 | 2.94 |
| C# | **2.27** | 2.28 | 2.29 |
| Common Lisp | **4.02** | 4.06 | 4.08 |
| Dart | **2.58** | 2.60 | 2.61 |
| Emacs Lisp | **9.55** | 9.67 | 9.69 |
| Erlang | **3.13** | 3.16 | 3.18 |
| Fortran | **4.28** | 4.34 | 4.38 |
| Go | **2.24** | 2.25 | 2.27 |
| Groovy | **3.62** | 3.66 | 3.68 |
| Haskell | **3.88** | 3.91 | 3.94 |
| Java | **2.43** | 2.45 | 2.45 |
| Pascal | **4.08** | 4.11 | 4.14 |
| Python | **3.32** | 3.35 | 3.36 |
| Mean | **3.69** | 3.73 | 3.74 |
| | Altered Train Data: Wikipedia | | |
| Eval/Method | SPLICE BM25 | SPLICE CONT | BASELINE |
| ArXiv | **5.64** | 5.65 | 5.73 |
| C | **2.65** | 2.67 | 2.71 |
| C++ | **2.98** | 3.01 | 3.07 |
| CUDA | **2.87** | 2.92 | 3.00 |
| C# | **2.22** | 2.24 | 2.29 |
| Common Lisp | **3.87** | 3.96 | 4.08 |
| Dart | **2.51** | 2.55 | 2.61 |
| Emacs Lisp | **9.38** | 9.45 | 9.63 |
| Erlang | **3.13** | 3.16 | 3.23 |
| Fortran | **4.23** | 4.32 | 4.49 |
| Go | **2.18** | 2.21 | 2.26 |
| Groovy | **3.49** | 3.55 | 3.67 |
| Haskell | **3.82** | 3.87 | 3.97 |
| Java | **2.39** | 2.41 | 2.46 |
| Pascal | 4.32 | **4.23** | 4.40 |
| Python | **3.26** | 3.30 | 3.37 |
| Mean | **3.68** | 3.72 | 3.81 |

Table 11: Perplexity results comparing different ways of organizing the same data. All runs started from the same $270M$ model with 2048 context and were trained for 32K context on a 50/50 mixture of RedPajama (organized in a standard way) and C code is organized in one of three ways. For details about training please refer to Section 3.1.

| | Altered Train Data: C | | |
| Eval/Method | SPLICE BM25 | BASELINE | RANDOM |
|---|---|---|---|
| ArXiv | **5.46** | 5.55 | 5.55 |
| C | **2.13** | 2.17 | 2.18 |
| C++ | **2.40** | 2.47 | 2.47 |
| CUDA | **2.22** | 2.33 | 2.33 |
| C# | **1.94** | 2.02 | 2.02 |
| Common Lisp | **2.99** | 3.20 | 3.18 |
| Dart | **2.14** | 2.27 | 2.27 |
| Emacs Lisp | **7.86** | 8.10 | 8.09 |
| Erlang | **2.67** | 2.77 | 2.77 |
| Fortran | **3.31** | 3.55 | 3.56 |
| Go | **1.91** | 2.00 | 2.00 |
| Groovy | **3.01** | 3.15 | 3.15 |
| Haskell | **3.20** | 3.37 | 3.37 |
| Java | **2.07** | 2.16 | 2.16 |
| Pascal | **3.49** | 3.62 | 3.64 |
| Python | **2.81** | 2.93 | 2.93 |
| Mean | **3.10** | 3.23 | 3.23 |

## C  ABLATIONS

### C.1  SPLICE PARAMETERS

There are two important design choices related to SPLICE. First, how many related documents are retrieved in each step (the parameter $k$ in Algorithm 1). Second, how the documents are ordered. Table 12 indicates that $k = 1$ is the best choice, though the differences are rather small. We found that changing the order of documents in training examples hardly matters. We use 'standard', as ordered by Algorithm 1, the reversed order, and random shuffling. We consider exploring these design choices to be an interesting future work direction.

Table 12: Ablation of SPLICE hyper-parameters. For each ablation, we have trained the same $270M$ parameter model using different data organization methods. Top-$k$ corresponds to the number of descendants chosen in the RETRIEVE$(d, k)$ step of the Algorithm 1. Reverse and shuffle correspond to the final order of examples $C$ returned by the Algorithm 1 (reverse – the order of documents in $C$ is reversed, shuffle – examples are shuffled.

| | | | Code | | | | |
| **Method** | **Top-$k$** | **Order** | C++ | Haskell | Python | CUDA | All |
|---|---|---|---|---|---|---|---|
| SPLICE BM25 | top 1 | standard | **2.38** | **3.20** | **2.82** | **2.23** | **2.93** |
| | | reverse | **2.38** | 3.21 | **2.82** | **2.23** | **2.93** |
| | top 2 | standard | 2.39 | 3.23 | 2.84 | 2.24 | 2.95 |
| | | reverse | 2.39 | 3.24 | 2.84 | 2.24 | 2.95 |
| | top 3 | standard | 2.39 | 3.24 | 2.84 | 2.25 | 2.97 |
| | | reverse | 2.39 | 3.25 | 2.84 | 2.25 | 2.96 |
| | | shuffle | 2.40 | 3.24 | 2.84 | 2.25 | 2.96 |

We also evaluated the influence of `BOS` and `EOS` tokens on the performance of trained models. As both REPO and SPLICE methods concatenate documents to create training examples, they effec-

tively slightly decrease the number of separating tokens compared to the BASELINE. However, in Appendix C.2 we included experiments showing that this has no impact on model performance.

## C.2 IMPORTANCE OF SEPARATING TOKENS

We also evaluate the influence of BOS and EOS tokens on the performance. To be more precise, in all setups training examples are separated by BOS and EOS tokens. As both REPO and SPLICE methods concatenate documents to create training examples, they effectively increase the average example length and decrease the number of separating tokens. To check whether those methods do not simply benefit from the reduction in the number of BOS and EOS tokens we have trained a model on data prepared similarly as in SPLICE, but instead of most matching documents $\text{RETRIEVE}(d, k)$ returned random documents from the dataset (sampling without replacement). The results are shown in Table 13. We note that the difference between the BASELINE and the random concatenation approach is small and the random concatenation approach does not result in significant perplexity gains.

Table 13: Perplexity evaluation of two methods of organizing the data. BASELINE – document equals training example. RANDOM – concatenate documents into examples of length bounded by 120k characters. Training examples are then fed into the model and separated by BOS and EOS tokens. The difference is negligible, which suggests that the extension of example length in RANDOM does not help the model to utilize extended context. Experiments were performed using 270M parameter model. We performed three runs on different subsets of C provide mean and standard deviation.

| Training Data | Method | arXiv | Code | | Code & arXiv |
|---|---|---|---|---|---|
| | | | Python | All | |
| C | RANDOM | $5.554 \pm 0.004$ | $\mathbf{2.931} \pm 0.003$ | $3.076 \pm 0.005$ | $3.231 \pm 0.005$ |
| | BASELINE | $\mathbf{5.550} \pm 0.002$ | $\mathbf{2.931} \pm 0.008$ | $\mathbf{3.073} \pm 0.006$ | $\mathbf{3.228} \pm 0.005$ |

## D KEY-VALUE RETRIEVAL TASK

Figure 4 shows how training on SPLICE organized data improves the performance on the key-value retrieval task proposed in (Liu et al., 2023). This is a zero-shot task in which a model is prompted with a JSON dictionary and asked to retrieve a value corresponding to a specified key. The structure of the input is showcased below.

```
Extract the value corresponding to the specified key in the JSON object below.

JSON data:
{"4ef217b7-6bc0-48c6-af35-2765f1e730f3": "068192b7-16b1-40e0-8495-61c63f979d50",
 "cd6b8bdc-bc6c-4490-acb4-bc187a2dccba": "7364a26e-289f-4968-93d3-b273e882bdee",
 "7d057372-4ab8-4811-8110-658c3f19fff4": "3ad075c5-b567-4201-85a7-cb31a0c91540",
 "c62e192d-45e6-4646-bb88-1529c73256c9": "f0411644-1f6d-42a6-8af8-f06da66efc77",
 "06134e93-e158-490e-a66c-8e3b98e12735": "50a26a36-d832-450c-8d6e-a4cc3d0ec0ab",
 "3286f978-4270-4b54-8bfa-540d7e0772e6": "075cc716-1836-4f90-9be3-53e3d4ec6585",
 "4701aa05-c523-4b89-9700-64ab9c37c537": "49d86354-74c4-4256-9b3a-35e6e2b80d00",
 "c8895805-e574-4f13-9fe5-89da1d8c4748": "cc91af7f-8509-4bdc-bad7-2646af68e6d2"}
 "4701aa05-c523-4b89-9700-64ab9c37c537":
```

## E PERPLEXITY IMPROVEMENTS

In Figure 5 we present perplexity improvements of SPLICE BM25 over BASELINE. Figure 6 shows evolution of SPLICE model perplexity during the training. We follow (Anthropic, 2023) and bucket perplexity by token positions in buckets of length $2^i$ up to 32768, and then average within the buckets. We average preplexity across arXiv, CUDA, Haskell and CommonCrawl datasets.

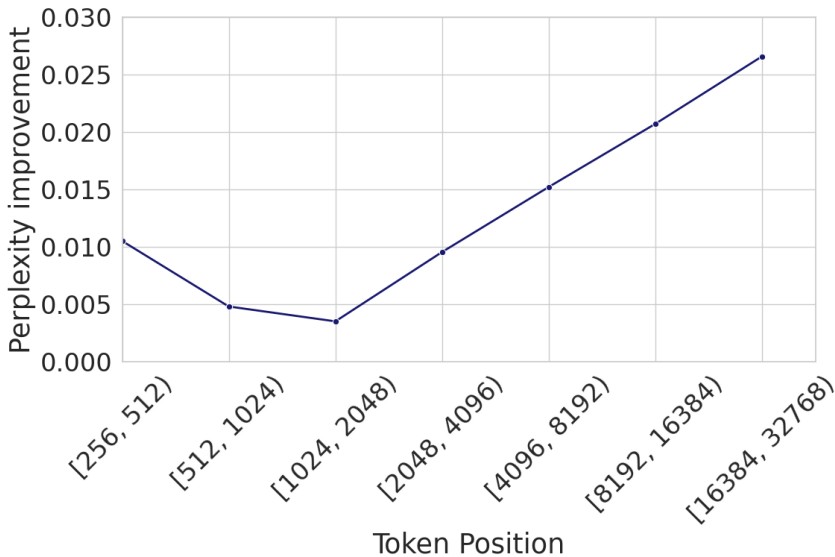

Figure 5: Perplexity improvement with SPLICE against the BASELINE of the final models (after 21k training steps). We bucket tokens by their positions in the document, and calculate the average. Each dot is the difference of the averages of the SPLICE and BASELINE models. We observe that SPLICE has smaller perplexity, and the improvements tend to be larger for tokens further in the document.

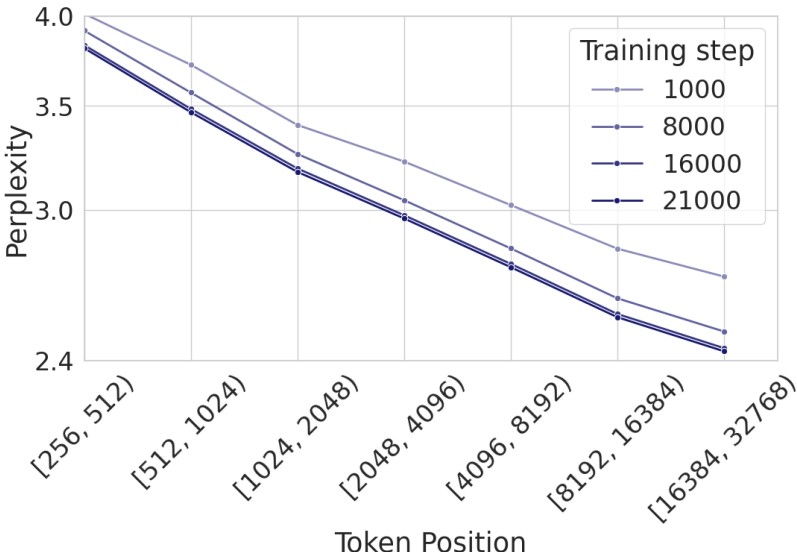

Figure 6: Evolution of the perplexity with SPLICE, as the model is trained on more tokens. See 5 for the difference with the baseline. As expected, SPLICE significantly improves perplexity for tokens whose positions are very distant in the sequence. Perplexity for more distant tokens improves more significantly compared to tokens in the beginning, early in the training.

## F    DATA PREPARATION

### F.1    EVALUATION DATA

We have taken a random subset of arXiv from Proof-pile. For StarCoder data, we have downloaded up to 64GB of each of the mentioned language subsets and performed a random 85/15 split for languages that we train on.

When evaluating the model we skip documents that are shorter than the model context and truncate documents that are longer than that. Table 14 shows the number of tokens over which the perplexity was calculated.

Table 14: Number of evaluation tokens in each of the considered datasets. For each context length $c$ we consider only documents having not less than $c$ tokens and extract the $c$ tokens prefix.

| Eval/Method | 16K | 32K | 64K |
|---|---|---|---|
| ArXiv | 16M | 16M | 16M |
| C | 16M | 16M | 16M |
| C++ | 16M | 16M | 16M |
| CUDA | 16M | 14M | 8M |
| C# | 16M | 16M | 16M |
| Common Lisp | 16M | 16M | 16M |
| Dart | 16M | 16M | 16M |
| Emacs Lisp | 16M | 15M | 9M |
| Erlang | 16M | 16M | 12M |
| Fortran | 16M | 16M | 16M |
| Go | 16M | 16M | 16M |
| Groovy | 11M | 4M | 2M |
| Haskell | 16M | 16M | 15M |
| Java | 16M | 16M | 16M |
| Pascal | 16M | 16M | 16M |
| Python | 16M | 16M | 16M |

## F.2 TRAIN DATA

The StackExchange data was taken from the Proof-pile. To prepare the code train data, we take the StarCoder train splits mentioned in Section F.1, shuffle them, group the documents by the repository (documents from the same repository occur one after another), and split them into smaller packs. We have decided to divide the dataset in this way to not weaken the REPO method (a simple random split would scatter repository files across the data packs). We also split repos larger than 25MB and filter out files that are longer than 30k characters. The reason behind repo splitting is to avoid the situation where one repository occupies a significant portion of the data pack. We have noticed repos containing as many as 40k files and files as long as 11M characters. The character filtering is consistent with our method as we aim to improve the performance in a scenario that lacks high-quality long-context data. For C# and Python, only one pack is used for BASELINE, REPO, and SPLICE to organize the data. For C we have performed a run on three packs and provided results and standard deviation in Table 2. For the $3B$ model we run the methods on several packs and concatenate the results into a single dataset.

## G TREC ACCURACY IMPROVEMENTS

Figure 7 show details about accuracy improvements on TREC when considering different numbers of in-context examples.

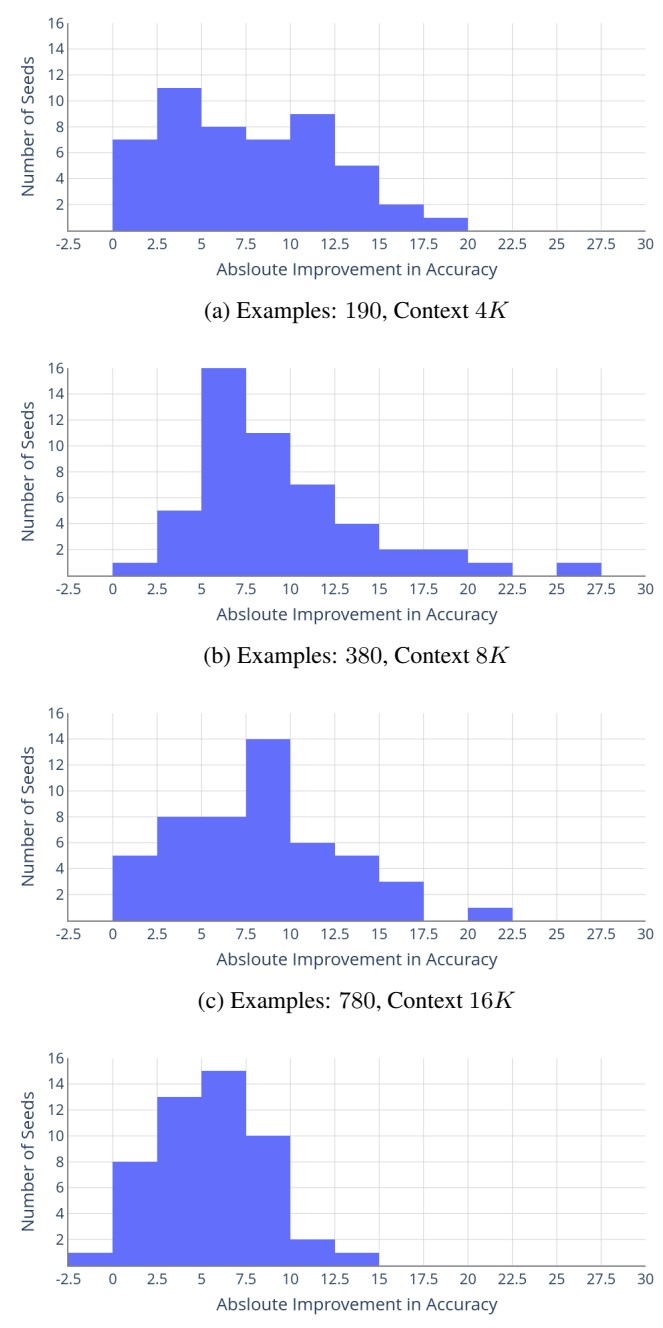

(a) Examples: 190, Context $4K$

(b) Examples: 380, Context $8K$

(c) Examples: 780, Context $16K$

(d) Examples: 1560, Context $32K$

Figure 7: Histograms of accuracy improvement of SPLICE BM25 over BASELINE on TREC question classification task. The results are obtained by comparing the accuracy on the test set of TREC of the 3B model trained with SPLICE to the model trained with default data preparation method (BASELINE) across 50 sets of in-context examples. Each set of in-context examples consists of elements randomly sampled (without replacement) from the training subset of TREC. Note that the model trained with SPLICE is almost always better than the BASELINE.

## H   FAISS PARAMETERS

Our experiments with SPLiCE ᴄᴏɴᴛ utilize Faiss (Johnson et al., 2017a) for fast approximate inner-product search. To be more precise we use the "IVF8192,Flat" index that we train on 262144 examples coming from the dataset.

## I   LARGER MODEL

We confirm our main results using larger 7B parameter models tuned from OpenLLaMA 7B v2. We present the in-context learning results on TREC (Li & Roth, 2002; Hovy et al., 2001) in Table 15, with scores averaged across 10 subsets of in-context examples. The 7B parameter models were tuned on a 50/25/25 mixture of RedPajama prepared in a standard way and StackExchange, C - prepared using SPLiCe.

Table 15: Few-shot performance on TREC. For TREC we average results across 10 sets of in-context examples.

| Task | Context length | 7B BASELINE | 7B SPLiCE |
|------|----------------|-------------|-----------|
| TREC | 32K | 75.6 ±4.4 | 79.4 ± 3.7 |
| | 16K | 74.0 ±2.6 | 79.0 ± 2.0 |
| | 8K | 72.1 ±3.5 | 76.2 ± 3.3 |
| | 4K | 66.9 ±4.3 | 69.7 ± 5.4 |

## J   LONGER CONTEXT

In table 16 we present the perplexity results for a 270M parameter models trained as described in Section 3 but with 2x larger context extension – context extended from 2K to 64K, instead of from 2K to 32K.

Table 16: Perplexity for training on a $50/50$ data mixture of RedPajama and C# code with longer 64K context.

| Training Data | Method | arXiv | Code | | | | Code & arXiv |
|---------------|--------|-------|---------|--------|------|-----|--------------|
| | | | Haskell | Python | CUDA | All | |
| C# | SPLiCE ʙᴍ25 | 4.86 | 2.60 | 2.66 | 2.32 | 2.75 | 2.88 |
| | SPLiCE ᴄᴏɴᴛ | 4.88 | 2.62 | 2.67 | 2.34 | 2.77 | 2.90 |
| | BASELINE | 5.01 | 2.79 | 2.82 | 2.51 | 2.94 | 3.07 |
| | REPO | 4.88 | 2.62 | 2.68 | 2.35 | 2.77 | 2.90 |

