# OpenReview forum: "Structured Packing in LLM Training Improves Long Context Utilization"
_ICLR.cc/2024/Conference — Submitted to ICLR 2024_

### Official Review · Reviewer_ixkc · 2023-10-31

**Soundness:** 3 good
**Presentation:** 3 good
**Contribution:** 2 fair
**Rating:** 3
**Confidence:** 4

**Summary:**

The paper presents a novel method called Structured Packing for Long Context (SPLICE) aimed at improving the context utilization in long-context Large Language Models (LCLMs). The authors identify that the lack of long-range semantic dependencies in typical training data hinders the effective utilization of context in LCLMs. To address this, they propose incorporating related documents more frequently into training inputs. By using BM25 to collate the most mutually relevant documents into a single training context, the authors demonstrate that SPLICE can enhance model performance across various tasks and can be used to train large models to better utilize long contexts. The method was validated by training a large 3B model and showed improvements in perplexity and better long-context performance on a benchmark key-value retrieval task.

**Strengths:**

The paper introduces an innovative method to improve the context utilization of LCLMs. The SPLICE approach, which involves structuring training data using BM25, is interesting and it can be applied to any textual data, making it more generally applicable.
The paper demonstrates that the application of SPLICE results in improvements in perplexity across various tasks.

**Weaknesses:**

1.  Using Lexical matching methods to concatenate the documents into a longer one is a very engineering technique and it is a straightforward solution to construct longer samples.

2. The experimental results are almost based on PPL, lacking experiments on real-world tasks. More experiments on benchmarks such as zeroScrolls[1] or L-Eval[2] to validate their models are needed. More extensive testing across a broader range of tasks and datasets would provide a more comprehensive evaluation of the method.

3. Presently, the prevalent strategies for training long context models involve the use of extensive conversations and literary works. A comparative analysis of SPLICE with these existing methodologies is thus a necessary step.

[1] ZeroSCROLLS: A Zero-Shot Benchmark for Long Text Understanding, 2023
[2] L-Eval: Instituting Standardized Evaluation for Long Context Language Models, 2023

**Questions:**

1. How does the choice of the BM25 method for document retrieval affect the model's performance? Would other document retrieval methods yield similar results?

2. How can we SPLICE  on very large pertaining corpus which usually has more than 400B tokens?

---

> ### Author Response · Authors · 2023-11-15
>
> We thank the Reviewer for a thoughtful review.
>
> We are not sure what point 1 of weaknesses means. We gently ask the reviewer for clarification.
>
> We agree that perplexity is not enough to fully evaluate the method. We now provide positive results for two datasets: TREC [1, 2] and Qasper [3, 4]. See details in the general answer and the revised pdf. In the general answer, we also provide new experiments with neural retrieval, which allow for efficient scaling (see the table below).
>
> We consider the techniques relying on extensive conversations (e.g. LongChat [5]) as complementary to our work. We note that SPLiCe can utilize data that already exists in vast quantities and can be easily applied to different types of text (like code, Wikipedia articles, stack exchange questions and answers, etc.). It is interesting to evaluate how both would interact, which we now include in the future work section on page 9.
>
> **Answers to the questions:**
>
> **Q1:** We now test our method using Contriever-MSMARCO [6]  instead of BM25 and observe that the results are very close. We attach the details in the general response and in the revised version of the paper (see Table 1 on page 5).
>
> **Q2:** SPLiCe using the Contriever-MSMARCO retrieval is scalable to large datasets. Namely, the dataset can be effectively queried using an approximate index (Faiss [7] in our case) which is a common technique, used in e.g. kNN-LM [8], RETRO [9].
> We have also performed an experiment to check whether SPLiCe will bring improvements when applied to a random subset of a larger set. For this, we have extracted 2GB from English Wikipedia and augmented it using SPLiCe. The results are presented in Table 3 on page 6. The outcome indicates that SPLiCe does not need to retrieve from the whole dataset to provide performance improvements. For example, one could split the training dataset into smaller parts and run SPLiCe independently for each of them reducing the complexity even further.
>
> [1] Li et. al. Learning Question Classifiers
> [2] Hovy et. al. Toward Semantics-Based Answer Pinpointing
> [3] Dasigi et. al. A Dataset of Information-Seeking Questions and Answers Anchored in Research Papers
> [4] Shaham et. al. SCROLLS: Standardized CompaRison Over Long Language Sequences
> [5] Li et. al. How Long Can Open-Source LLMs Truly Promise on Context Length?
> [6] Izacard et.al. Unsupervised Dense Information Retrieval with Contrastive Learning
> [7] Johnson et. al. Billion-scale similarity search with GPUs
> [8] Khandelwal et. al. Generalization through Memorization: Nearest Neighbor Language Models
> [9]  Borgeaud et. al. Improving language models by retrieving from trillions of tokens

---

> ### Author Response · Authors · 2023-11-20
>
> We hope that our explanations and additional results show the potential of SPLiCe. In particular, our evaluation on additional downstream tasks along with results regarding Contriever-MSMARCO and Faiss replacing BM25.
>
> Considering the approaching rebuttal deadline, we ask the Reviewer for comments and a score increase in case our findings meet the requirements.

---

### Official Review · Reviewer_wDkv · 2023-11-02

**Soundness:** 2 fair
**Presentation:** 1 poor
**Contribution:** 2 fair
**Rating:** 3
**Confidence:** 3

**Summary:**

The paper proposes structured packing for long context (SPLiCE) that constructs long context training examples by retrieving relevant documents using BM25. After experiments on a small language model with different datasets and configurations, SPLiCE is applied to large-scale language models.

**Strengths:**

The main idea of constructing better training examples makes sense. SPLiCE is not too complicated and does not require expensive overhead or external models by relying on BM25.

**Weaknesses:**

Considering the additionally introduced complexity (though the SPLiCE algorithm is simple), the performance improvement looks very marginal, especially for large-scale models.
Only the part of the training is replaced with SPLiCE from the random baseline. That might be one of the reasons for marginal improvement, but it also implies that SpLiCE is not a standalone solution that can completely replace the existing training algorithm.

Language modeling perplexity is the main evaluation metric. Comparing performance on other NLP downstream tasks that require long context modeling might be better to evaluate the effectiveness of the proposed method.

**Questions:**

I raised several concerns about why SPLiCE is not sufficient (or at least not fully validated) as it is. Could you address them?

I guess the number of neighbors for each document is skewed, meaning that there exists hub documents. In that case, although a root document is randomly sampled from the document corpus, the retrieved documents are not uniformly distributed in terms of their likelihood and order. Couldn't this be a problem that may result in an imbalance in training?

Packed documents are unnatural and different from contiguous documents. Is there any way to alleviate this issue?

As expected, using related documents in a long context is better than the random baseline. However, any design choices (top-k, order, or even REPO vs. SPLiCE) give clear differences.
In particular, top-1 is the best, and in that case, BFS is the same as DFS.

Why is Table 1 required? Table 2 fully covers Table 1.

The structure of the paper can be improved. For example, it is awkward that Section 4 also includes experiments while the title of Section 3 is experiments. Also, multiple Figures and Tables can be merged to spare some space for more extensive experiments or discussion.

---

> ### Author Response · Authors · 2023-11-15
>
> We thank the Reviewer for their insightful feedback.
>
>
> We note that SPLiCe is a stand-alone solution. Namely, it could be used for the whole training. We use it in fine-tuning due to computational constraints. We also note that it is fairly standard to train on a short context for the majority of training steps and extend it in the final phase, see e.g. CodeLlama [1]. We believe that the performance gains validate the usage of the algorithm, taking into account that it is simple and its computational overhead is negligible compared to the training costs.
>
> We totally agree with the request for NLP evaluation. We now provide results for TREC [2, 3] and Qasper [4, 5] datasets, which confirm the usefulness of the SPLiCe training. Please refer to the results in the general answer.
>
> In what follows, we address the Reviewer’s questions:
> * Regarding the imbalance. We use each document only once. A document that is used in training is removed (see the last lines of Alg 1). This design choice ensures that for one epoch the whole dataset is seen. What is more, we mix the SPLiCe prepared data with data prepared in a standard way, which should prevent the model from overfitting to the structure of SPLiCe constructed examples. In our setup, we have not noted training instabilities.  We agree that there is some subtlety here and open questions, which we now note in the further research section (see page 9).
> * We agree that there is some risk in using ‘unnatural’ packed documents. To alleviate this we train on a mixture of data prepared in the standard way and SPLiCe (which is now explained in Section 3.1). Based on [1, 6] we think that using data mixture is enough to alleviate false biases.
> * SPLiCe is a general framework for constructing long context data using short documents. For example, for k=1 and the retriever returning elements from the closest subdirectory in the repository we get the Repo method [7]. With larger k and reversed order we get the data that resembles inputs to retrieval augmented models (e.g.  REALM [8]). We believe that leaving the possibility for other choices paves the way for future research and is of use to the community.
> * Thank you for pointing out several exposition improvement areas. We have made multiple changes towards this goal (e.g. the mentioned split of the results in Table 1 and Table 2 and the titles) and we will gladly accept any further suggestions.
>
> [1] Rozière et. al. Code Llama: Open Foundation Models for Code
> [2] Li et. al. Learning Question Classifiers
> [3] Hovy et. al. Toward Semantics-Based Answer Pinpointing
> [4] Dasigi et. al. A Dataset of Information-Seeking Questions and Answers Anchored in Research Papers
> [5] Shaham et. al. SCROLLS: Standardized CompaRison Over Long Language Sequences
> [6] Ouyang et. al. ”Training language models to follow instructions with human feedback
> [7] Wu et. al. Memorizing Transformers
> [8] Guu et. al.  REALM: Retrieval-Augmented Language Model Pre-Training

---

> ### Author Response · Authors · 2023-11-20
>
> Using the additional time, we managed to evaluate our method using larger 7B parameter models, longer context and more downstream tasks (the details are present in general response). We hope that our additional results and explanations showcase the potential of SPLiCe.
>
> Considering the approaching rebuttal deadline, we ask the Reviewer for comments and a score increase in case our findings meet the requirements.

---

### Official Review · Reviewer_21Pg · 2023-11-04

**Soundness:** 3 good
**Presentation:** 3 good
**Contribution:** 3 good
**Rating:** 6
**Confidence:** 4

**Summary:**

The paper proposes SPLICE, a similarity-based approach of grouping documents into pretraining examples to training better long context language models. For each example, the method starts with a single document and uses a BM25 retriever to include more relevant documents in the example in a BFS fashion.

When applied to training a 270M model, the method outperforms the random baseline on both text and code perplexity. The model is also on par with the REPO method which relies on knowledge of the corpus structure. When used to train a 3B model, the method also outperforms the baseline on both perplexity and the key-value retrieval task. Ablation studies are included to analyze the impact of hyperparameters.

**Strengths:**

- The method is simple yet effective and can be easily applied to different scenarios.
- Reproducibility: The authors attach the source code, which is great. Please also release the code if the paper is accepted.
- Clarity: the paper is well written and easy to understand.
- Significance: the significance of the paper is okay.

**Weaknesses:**

- The effectiveness of the method is only validated on language modeling and the key-value retrieval task. This does not guarantee the resultant model is stronger on realistic use cases. To test the usefulness of SPLICE, I would highly recommend comparing the models on more challenging and realistic long-context downstream tasks such as Quality and Squality.
- It would be great if the method is tested on more settings: Use a neural retriever in addition to BM25, go beyond 3B and 32K, etc.
- Novelty: The main idea is quite similar to many existing methods like the ones discussed in the paper (e.g. retro). However, I don't think the paper should be rejected only because of this.

**Questions:**

- 3.4 “On the code evaluations, the improvements are small.” - Why say so? The average improvement on code datasets is 0.0625 and the improvement on arxiv is 0.07. The improvements seem to be similar.
- 4.2: “The perplexity difference is larger for tokens further in the document” - I might misunderstand something, but it seems the improvement is also large at the start? (Figure 3)
- Typo: 3.2 “Moreover, even thorough this method uses only the code data”
- In the abstract “Our results indicate that SPLICE enhances model performance across various task” - The context of this sentence is the 270M model. It is in fact only tested on one task: language modeling (though there are different datasets). You might want to rephrase to reduce confusion.

---

> ### Author Response · Authors · 2023-11-15
>
> We thank the Reviewer for an encouraging review.
>
> We now provide results showing that the SPLiCe training is useful for downstream NLP tasks: TREC [1, 2] and Qasper [3, 4]. Likewise, we are happy to report results for SPLiCe using Contriever-MSMARCO [5]. Please see the details in the general answer and the revised pdf and let us know if this is satisfactory.
>
> Due to computational constraints, we have not been able yet to go beyond the current size/context length but we hope to provide results for the 7B/32K model and 270M/64K model before the rebuttal deadline.
>
>
>
> **Regarding questions:**
>
> **Q1:** We acknowledge that this statement is confusing. We have rephrased it as follows: "Despite the non-code nature of StackExchange we still note perplexity improvements on code. Unsurprisingly, the perplexity on code is not as good as when training directly on code data.”.
>
> **Q2:** We apologize, but a mistake has slipped through, and we have unintentionally provided results of the 3B parameter model that was trained with a context length smaller than 32K. We corrected that in the revised version, providing the corresponding perplexity improvement results that now match the description. We also re-evaluated the model on the kv retrieval task and added the comment about improved performance. See the revised pdf, page 8 and 20.
>
> **Q3:** We thank you for spotting the typo. We have fixed it and merged sections 3.2 and 3.3 into one.
>
> **Q4:** Thank you for the suggestion. We have rephrased this part as follows: “Our results indicate that SPLiCe enhances model performance and can be used to train large models to utilize long contexts better.”
>
>
>
> [1] Li et. al. Learning Question Classifiers
> [2] Hovy et. al. Toward Semantics-Based Answer Pinpointing
> [3] Dasigi et. al. A Dataset of Information-Seeking Questions and Answers Anchored in Research Papers
> [4] Shaham et. al. SCROLLS: Standardized CompaRison Over Long Language Sequences
> [5] Izacard et.al. Unsupervised Dense Information Retrieval with Contrastive Learning

---

> ### Author Response · Authors · 2023-11-20
>
> Using additional time, we extended our research to 7B parameter models with 32K context and 270M parameter models with 64K context length. We add additional results to the general response. We hope that those findings along with the ones presented in our previous responses showcase the potential of SPLiCe. In particular our previous results have tested the method on two additional downstream tasks.
>
> Considering the approaching rebuttal deadline, we ask the Reviewer for comment and a score increase in case our findings meet the requirements.

---

### Author Response · Authors · 2023-11-15

We thank the Reviewers for their time, effort, and appreciation of our work, particularly in the areas of simplicity (21Pg, wDkv), clarity (21Pg), and generality (ixkc).

At the same time, we value constructive critique and acknowledge pointed shortcomings. In our opinion, the quality of our work has been substantially improved, and we addressed multiple weaknesses. We have updated the pdf accordingly, and highlighted in blue the notable changes vs the original version. The major improvements are:

* we showed that the perplexity improvements transfer to downstream tasks,
* we showed that the neural retriever along with an efficient approximate kNN search could be used as a replacement for BM25,
* we made numerous improvements to the presentation and increased the number of runs for some experiments. In addition to this:
    * we have merged the Repo and SPLiCe code evaluation sections
    * we have moved some of the perplexity results to the appendix
* we provided results for other RoPE-based context extension methods (see Section 3.4 on page 6)


## Downstream tasks improvements

First, we test the in-context learning ability of our 3B parameter models using the TREC [1,2] questions classification task for different context lengths. Notably, the SPLiCe model outperforms the baseline even when using 4 times shorter inputs. In greater detail, for each context length, we sample 50 subsets of in-context examples from the train set. For each of them, we compute the performance on the test set and average the results. For details, see Figure 7 in the revised pdf, as it shows that for virtually all choices of in-context examples, the SPLiCe model is strictly better. The table below reports the average over the 50 subsets.  Additionally, we test the question-answering ability of our 3B parameter models using Qasper [3, 4] and report improvements.

In the table below, we present the results of new tests on the TREC and Qasper tasks.

| Task          | Context length | 3B Baseline | 3B SPLiCe |
|---------------|-----------------|-----------------------|--------------|
| **TREC**      | 32K             | 73.9 $\pm$ 3.9        | 79.3 $\pm$ 2.9|
|               | 16K             | 68.9 $\pm$ 5.9        | 76.9 $\pm$ 3.1|
|               | 8K              | 66.5 $\pm$ 6.2        | 75.8 $\pm$ 3.5|
|               | 4K              | 65.1 $\pm$ 6.1        | 72.8 $\pm$ 4.9|
|               |                 |                       |              |
| **Qasper F1** | 32K             | 23.1                  | 23.9         |
|               | 4K              | 18.6                  | 18.7         |


## Contriever results
For Contriever-MSMARCO [5] (denoted as SPLiCe CONT) we embed the first 512 tokens and utilize Faiss [6] for fast approximate inner-product search. We present the details in Appendix H. In a separate comment we attach the updated table.

[1] Li et. al. Learning Question Classifiers
[2] Hovy et. al. Toward Semantics-Based Answer Pinpointing
[3] Dasigi et. al. A Dataset of Information-Seeking Questions and Answers Anchored in Research Papers
[4] Shaham et. al. SCROLLS: Standardized CompaRison Over Long Language Sequences
[5] Izacard et. al. Unsupervised Dense Information Retrieval with Contrastive Learning
[6] Johnson et. al. Billion-scale similarity search with GPUs

---

> ### Author Response · Authors · 2023-11-15
>
> ### Table: Training Methods and Perplexity Results for Various Programming Languages
> We train a 270M parameter model on a 50/50 mixture of the RedPajama data (organized in a standardway) and code data (organized in one of four ways: SPLiCe BM25, SPLiCe Contriever-MSMARCO, Repo, Baseline – standard). We evaluate the perplexity of the models on arXiv and various subsets of StarCoder. The boldface denotes the best result within the training mixture. Note that SPLiCe is generally on par or better than Repo and beats the Baseline by a large margin. For detailed results, see Appendix B.
>
> | **Training Data** | **Method** | **arXiv** | **Code (Haskell)** | **Code (Python)** | **Code (CUDA)** | **Code (All)** | **Code & arXiv** |
> |-------------------|------------|-----------|---------------------|-------------------|------------------|-----------------|------------------|
> | **C**             | SPLiCe BM25              | **5.46**   | **3.20**           | **2.81**         | **2.22**         | **2.94**        | **3.10**         |
> |                   | SPLiCe CONT | 5.48       | 3.22              | 2.82             | 2.23             | 2.96            | 3.11             |
> |                   | Baseline                | 5.55       | 3.37              | 2.93             | 2.33             | 3.07            | 3.23             |
> |                   | Repo                    | 5.47       | 3.22              | 2.83             | 2.24             | 2.96            | 3.12             |
> | **C#**            | SPLiCe BM25              | **5.52**   | **3.33**           | **2.90**         | **2.46**         | **3.11**        | **3.26**         |
> |                   | SPLiCe CONT | 5.53       | 3.35              | 2.91             | 2.48             | 3.12            | 3.27             |
> |                   | Baseline                | 5.65       | 3.58              | 3.07             | 2.65             | 3.31            | 3.46             |
> |                   | Repo                    | 5.53       | 3.35              | 2.91             | 2.49             | 3.12            | 3.27             |
> | **Python**        | SPLiCe BM25              | **5.47**   | **3.25**           | **2.53**         | **2.41**         | **3.02**        | **3.17**         |
> |                   | SPLiCe CONT | 5.49       | 3.28              | **2.53**         | 2.43             | 3.03            | 3.19             |
> |                   | Baseline                | 5.57       | 3.46              | 2.62             | 2.56             | 3.17            | 3.32             |
> |                   | Repo                    | 5.48       | 3.27              | 2.54             | 2.44             | 3.03            | 3.18             |

---

> ### Author Response · Authors · 2023-11-20
>
> ## 7B parameter model
> We have managed to tune the 7B parameter OpenLLaMA v2 model for 32K context length using 2B tokens 50/25/25 mixture of Redpajama prepared in standard way and StackExchange, C - prepared using SPLiCe. We compared the performance on TREC of the SPLiCe model to the model that was trained on data prepared in a standard way. The results are available in the table below. For each context length, we average the results across $10$ sets of in-context examples.
> | Context length | 7B Baseline | 7B SPLiCe |
> |----------------|-----------------------|--------------|
> | 32K            | 75.6 $\pm$ 4.4        | 79.4 $\pm$ 3.7 |
> | 16K            | 74.0 $\pm$ 2.6        | 79.0 $\pm$ 2.0 |
> | 8K             | 72.1 $\pm$ 3.5        | 76.2 $\pm$ 3.3 |
> | 4K             | 66.9 $\pm$ 4.3        | 69.7 $\pm$ 5.4 |
>
> ## 64K context length
> We have managed to tune 270M parameter models for 64K context following the approach presented in the paper. The evaluation results are available in the table below.
> | **Training Data** | **Method** | **arXiv** | **Haskell** | **Python** | **CUDA** | **All** | **Code & arXiv** |
> | ----------------- | ---------- | --------- | ----------- | --------- | ------- | ------- | ---------------- |
> | C#                | SPLiCe BM25    | 4.86      | 2.60        | 2.66      | 2.32    | 2.75    | 2.88             |
> |                   | SPLiCe CONT | 4.88  | 2.62        | 2.67      | 2.34    | 2.77    | 2.90             |
> |                   | Baseline   | 5.01      | 2.79        | 2.82      | 2.51    | 2.94    | 3.07             |
> |                   | Repo       | 4.88      | 2.62        | 2.68      | 2.35    | 2.77    | 2.90             |
>
> We plan to add those additional results to the pdf before the rebuttal deadline.

---

> > ### Author Response · Authors · 2023-11-22
> > **End of time**
> >
> > Dear Reviewers.
> >
> > We made a substantial effort to increase the quality before and during the rebuttal. We appreciate your work. We did our best to address your suggestions and concerns. However, at the same time, we feel discouraged by the lack of discussion.
> >
> > We kindly ask you to acknowledge reading our rebuttals, at the very least.

---

> ### Author Response · Authors · 2023-11-23
>
> Due to the approaching rebuttal deadline, we have uploaded the PDF version without color-marked differences. We have also added the 7B  parameter model results and results with the 64K context for 270M parameter models. Those results were previously stated in our comments.

---

### Meta-Review · Area_Chair_GCkZ · 2024-01-05

**Metareview:**

The authors explore how to select higher utility content to include in (long-context) LLM context windows, proposing a method for retrieving (and assembling) groups of similar documents to pack into the context, referred to as "Structured Packing for Long Context" (SPLiCe). The conceptual motivation is that widely-used training LLM training sources frequently lack long-range dependencies and that incorporating related documents more frequently into the context window can improve performance (instead of "randomly" packing documents into the context window). Operationally, they perform this using the retrieval scoring function as a basis for a hierarchical clustering-like procedure and pack the context window by flattening the resulting tree via a tree traversal. An empirical study is performed on 270M and 3B models to show that SPLiCe outperforms a random packing baseline on both text and code perplexity and is competitive with a selection procedure that uses meta-data associated with directory structure. Additional experiments are performed to demonstrate improvements on the key-value retrieval task for the 3B model setting and hyperparameter sensitivity in multiple settings.

Consensus strengths identified by reviewers regarding this submission include:
- The motivation (better packing of context windows) is useful and likely of interest to many proposed LLM approaches. The proposed method is intuitively appealing, conceptually simple, and easily generalizes to many settings.
- The empirical results demonstrate that applying SPLiCe results in improvements in perplexity and key-value retrieval across multiple tasks. Additionally, during rebuttal, they expanded these findings to include performance improvements for 'downstream' NLP tasks (e.g., question classification). I believe the results are convincing that SPLiCe is an improvement over a 'random packing' approach.
- From the perspective of understanding the SPLiCe technical details, the paper is well-written and easy to understand.

Conversely, consensus limitations included:
- In the original submission, evaluation was based on perplexity and key-value retrieval. While they expanded this during rebuttal, the resulting modified submission doesn't have a lot of discussion regarding the new results and seems to require further 'deep dive' analysis to understand the precise claims/ramifications of the new experiments.
- In the original submission, the implementation was limited to BM25 for retrieval, smaller LLMs (e.g., 3B being the largest), and relatively smaller context windows. This is problematic as it is required to evaluate the longer-term value of SPLiCe as language models and context windows grow. This was somewhat addressed during rebuttal by exploring more configurations (e.g., context window size, neural retrieval, larger LLM). However, as with the previous concern, the new results are presented without sufficient discussion and 'deep dive' to develop a precise proved claim (i.e., is there a relationship between context window size and margin of improvement? [it seems so, but this isn't investigated in detail]).
- There were remaining concerns regarding sufficient contextualization with respect to related work to establish precise novelty claims. Specifically, is a 'random packing' baseline the best starting point for comparision?

Overall, I believe this is a good research direction and SPLiCe is a simple and general method that offers improvements over a 'random packing' baseline. I believe the original submission didn't have a strong evaluation, while the additional experiments provided during rebuttal make this a more convincing paper. That being said, I still believe that additional discussion of the results is necessary to demonstrate trends in the SPLiCe configuration space to better understand the method. It currently reads as a set of observations without establishing the strengths/weaknesses of different aspects. Additionally, it seems that there are other heuristic baselines that may be better than a random one -- or at least this should be discussed. This is an interesting direction, but requires more work before publication.

**Justification For Why Not Higher Score:**

The papers most interesting results were presented during rebuttal and still require more exploration to characterize the performance of SPLicE well.

**Justification For Why Not Lower Score:**

N/A

---

### Decision · Program_Chairs · 2024-01-16

Reject